# Examining the Relationship between Circulating CD4− CD8− Double-Negative T Cells and Outcomes of Immuno-Checkpoint Inhibitor Therapy—Looking for Biomarkers and Therapeutic Targets in Metastatic Melanoma

**DOI:** 10.3390/cells10020406

**Published:** 2021-02-16

**Authors:** Sabino Strippoli, Annarita Fanizzi, Antonio Negri, Davide Quaresmini, Annalisa Nardone, Andrea Armenio, Angela Monica Sciacovelli, Raffaella Massafra, Ivana De Risi, Giacoma De Tullio, Anna Albano, Michele Guida

**Affiliations:** 1Rare Tumors and Melanoma Unit, IRCCS Istituto Tumori Giovanni Paolo II, 70124 Bari, Italy; strippoli.sabino@libero.it (S.S.); davide.quaresmini@hotmail.it (D.Q.); angelamonicasciacovelli@gmail.com (A.M.S.); ivana.derisi@gmail.com (I.D.R.); annaalbano83@libero.it (A.A.); 2Radiotherapy Unit, IRCCS Istituto Tumori Giovanni Paolo II, 70124 Bari, Italy; annarita.fanizzi.af@gmail.com (A.F.); nardone.annalisa@gmail.com (A.N.); massafraraffaella@gmail.com (R.M.); 3Unit of Hematology and Cell Therapy, IRCCS Istituto Tumori Giovanni Paolo II, 70124 Bari, Italy; negriantonio@hotmail.it (A.N.); minadetullio@hotmail.it (G.D.T.); 4Department of Plastic Surgery, IRCCS Istituto Tumori Giovanni Paolo II, 70124 Bari, Italy; andreaarmenio@hotmail.com

**Keywords:** double negative T cells, checkpoint inhibitors, melanoma, immunotherapy resistance

## Abstract

Background: The role of circulating CD4**−**/CD8**−** double-negative T cells (DNTs) in the immune response to melanoma is poorly understood, as are the effects of checkpoint inhibitors on T cell subpopulations. Methods: We performed a basal and longitudinal assessment of circulating immune cells, including DNTs, in metastatic melanoma patients treated with checkpoint blockade in a single-center cohort, and examined the correlations levels of immune cells with clinical features and therapy outcomes. Results: Sixty-eight patients (48 ipilimumab, 20 PD1 inhibitors) were enrolled in the study. Our analysis indicated that better outcomes were associated with normal LDH, fewer than three metastatic sites, an ECOG performance status of 0, M1a stage, lower WBC and a higher lymphocyte count. The increase in lymphocyte count and decrease of DNTs were significantly associated with the achievement of an overall response. The median value of DNT decreased while the CD4+ and NK cells increased in patients that responded to treatment compare to those who did not respond to treatment. Conclusions: DNT cells change during treatment with checkpoint inhibitors and may be adept at sensing the immune response to melanoma. The complementary variation of DNT cells with respect to CD4+ and other immune actors may improve the reliability of lymphocyte assessment. Further investigation of DNT as a potential target in checkpoint inhibitor resistant melanoma is warranted.

## 1. Introduction

Predictive markers for immunotherapy are needed to improve patient selection and avoid toxicity and wasted treatment resources. Despite extensive research, a single biomarker that is able to discriminate between response and resistance to immune checkpoint inhibitors is yet to be identified. It is conceivable that the right combination of clinical features, serum factors, blood cells, cancer genetic signature, and tumor microenvironment could be used to construct an overall definition of the ideally responsive patient. Among these variables, the most practical one would be the assessment of parameters from peripheral blood sampling, which ensures minimal invasiveness and fast and dynamic monitoring of these parameters over time. In stage IV melanoma, where immunotherapy has been a well-established backbone of treatment since 2011 [1], the serum LDH level and the absolute lymphocyte count (ALC) are widely used and reliable blood predictive markers of response to both ipilimumab and PD1 inhibitors [2,3,4,5,6,7]. However, these parameters are still weak because of their high prognostic rather than predictive value, as well as their low specificity. LDH is also influenced by haemolysis and inflammatory injuries, and lymphocytes account for a heterogeneous spectrum of cell populations with different, sometimes opposite, functions. To narrow the selection of the ideal biomarker, a deeper analysis is required regarding the role of the diverse immune actors in the anti-melanoma response, and the effects of CTLA4 and PD1-inhibitors on different T cell subpopulations. Among CD3+ cells, a potential point of interest is aroused by a small cell subset characterized by the expression of either αβ or γδ T cell receptors (TCR) and the lack of mature surface T markers, such as CD4, CD8, CD56, which are known as double-negative T cells (DNTs). DNTs account for about 3% of circulating T cells and display a phenotypic plasticity depicted as a two-faced Janus attitude by exerting tolerogenic activity as regulatory cells useful in preventing graft versus host disease (GvHD) and cytotoxic behavior exploited in graft versus leukemia and solid neoplasm [8,9,10]. DNTs could be relevant in the homeostatic organization involved in the cooperative control of the magnitude of immune responses [9]. In the onco-haematology field, interest in DNTs stems from their use in adoptive T cell therapy, which is suggested by their safe infusion and their anti-tumor effects enhanced by check-point inhibitors in xerograft models [11,12,13,14]. However, the function of DNTs remains elusive and some controversial issues still need to be resolved. In particular, the biological meaning of the DNT count in peripheral blood is unknown, as it seems to show opposite behaviors in different settings. In allogeneic transplantation, circulating DNTs are inversely related to the risk and severity of acute and chronic GvHD [15]. In connective tissue diseases, such as systemic lupus erythematosus, DNTs were investigated as potential circulating biomarkers of kidney dysfunction [16]. Notably, a DNT clone presenting an MHC class I complex restricted for gp100 peptide was isolated from the peripheral blood of a melanoma patient and exerted an in vitro cytotoxicity against melanoma cells [17]. Moreover, in a previous study [18], our group reported a significant decrease of circulating DNTs in a metastatic melanoma (MM) population as compared both with lymphoma patients and healthy controls, and we found a positive correlation between DNT levels and ECOG performance status in MM patients. The presence of DNTs has been demonstrated within lymph node metastases of melanoma in the shape of tolerogenic T cells [19], and their accretion has been described in a serial biopsy of melanoma metastases in patients treated with BRAF and MEK inhibitors [20]. The relationship between the burden of circulating DNTs and outcomes of checkpoint blockade therapy has not yet been investigated. Thus, we performed a basal and longitudinal assessment of various immune cell populations, including DNT cells, from peripheral blood samples of metastatic melanoma patients treated with checkpoint blockades in a single-center cohort, with the aim of verifying the potential role of DNTs as a predictive marker of response to Ipilimumab or PD1 inhibitors and how the DNT count is influenced by these treatments.

## 2. Materials and Methods

### 2.1. Patients, Treatment and Assessment

We built an observational cohort study by recruiting prospectively patients with stage IV melanoma treated with ipilimumab or PD1 inhibitors (nivolumab or pembrolizumab). Patients were treated according to the standard dose and schedule of ipilimumab, nivolumab or pembrolizumab. Ipilimumab was administered at 3 mg/kg every 3 weeks for four infusions. Pembrolizumab and nivolumab were administered at 2 mg/kg every 3 weeks and 3 mg/kg every 2 weeks until progression, respectively. Peripheral blood samples were collected at baseline (the day of the first cycle) and when the first radiological assessment was available. Immune Response Criteria for Solid Tumors (iRECIST) [21] were used to evaluate the radiological assessment, which was performed by computer tomography(CT) scan every three months. We recorded the following clinical features of patients: Age, sex, melanoma type, TNM stage according to AJCC 8th edition [22], disease-free survival (DFS), number of metastatic sites, treatments, response to therapy, ECOG, and LDH value before and during immunotherapy. Follow up continued until patients died or until the final update in October 2020. The study was approved by the local Ethics Committee of Istituto Tumori “Giovanni Paolo II” of Bari (prot. no 394/EC of 9 October 2012) and was conducted in accordance with the international standards of good clinical practice.

### 2.2. Flow Cytometer Evaluation

Samples of venous blood (30 mL) were drawn into ethylene diamine tetra acetic acid (EDTA) tubes, carried by hand to the laboratory, and immediately processed for flow cytometer evaluation. To block unspecific binding of antibodies, cells were incubated with 2.5 μg/mL mouse IgG (Sigma-Aldrich, St. Louis, MO, USA, Cat-no: I8765-10MG) diluted in phosphate buffered saline (PBS, Sigma-Aldrich, St. Louis, MO, USA, Ref-no: 14190-094) for 15 min on ice. Next, the cells were stained with fluorochrome-labeled monoclonal antibodies.

The following anti-human antibodies for staining of cell surface markers were used: CD3-ECD (clone UCHT1), CD4-FITC or -PECy7 (clone SFCI12TAD11), CD8-FITC (clone B9.11), CD56-APC (clone N901), TCR PAN αβ-PE (clone IP26A), TCR PAN γδ-FITC (clone IMMU 510), CD45-PECy5.5 or -ECD (clone J.33), CD20-PECy5.5 (clone B9E9) were purchased from Beckman Coulter Inc. Brea, CA, USA. Data were acquired by an 8-color NAVIOS^®^ flow cytometer (Beckman Coulter Inc., Brea, CA, USA) and analyzed using Kaluza Analysis Software (Beckman Coulter Inc. Brea, USA.). We acquired 100,000 for each sample.

### 2.3. Statistical Analysis

A Chi-square test was used to evaluate significant associations between overall survival (OS) (months) and progression-free survival (PFS) (months) and categorical clinical variables including ECOG, LDH (≥ or < ULN), number of metastatic sites, BRAF mutational status, and metastatic stage.

The Wilcoxon Mann–Whitney test was used to evaluate the significant association between Overall Response Rate (defined as Complete Response/Partial Response vs. Progressive Disease) and a categorical clinical variable. Stable disease was excluded from this analysis.

Finally, the Pearson correlation coefficient was used to evaluate the association between OS (months) or PFS (months) and the numeric variables age, and T cell values.

A result was considered significant when the *p*-value was less than 0.05 or 0.10. All calculations were performed using SPSS statistical software.

## 3. Results

### 3.1. Therapy Outcomes and Correlations with Clinical Features

Between December 2013 and July 2016, 68 patients were enrolled in the study, 48 of whom were treated with ipilimumab and 20 of whom were treated with PD1 inhibitors. The clinical features of the patients are reported in Table 1. The overall response rate (ORR) in the entire population was 25% (10% complete response and 15% partial response) and the disease control rate (DCR: Complete or partial response and stable disease for longer than 6 months) was 37%. In the ipilimumab group, the ORR was 19% and the DCR was 29%, while in the anti-PD1 group the ORR was 40% and the DCR was 55%. For the entire cohort, the median PFS was 3 months (range 1–80). In the ipilimumab-treated group, the median PFS was also 3 months, with two patients who did not progress at 70 and 80 months, while for those treated with PD1 blockade, the median PFS was 8 months (range 1–63), with three patients who did not progress at 57, 60 and 63 months (Figure 1a). For patients in the ipilimumab group the median OS was 8 months (2–80), with nine (19%) patients still alive. The median OS was 8 months (2–63) in the PD1 inhibitor-treated population, with three patients still alive (Figure 1b).

The statistical analysis showed that the ORR was significantly higher for patients with an ECOG performance status of 0, fewer than three metastatic sites < 3 and age superior to the median value (Table 2). M1a stage and normal LDH were also significantly correlated with ORR for patient treated with ipilimumab (*p* < 0.05).

PFS was significantly correlated with M1a stage, better ECOG performance status, number of metastatic sites < 3 and normal LDH in the entire population (Table 3).

Overall survival was significantly correlated with increased age (r = 0.25, *p* < 0.10), and was significantly higher in patients with melanoma at stage M1a stage, NRAS mutation, better ECOG performance status, an LDH below the ULN, and fewer than three metastatic sites (Table 4).

### 3.2. Correlation Between the Baseline Value of T Cells, Clinical Features and Therapy Outcomes

At baseline, a higher absolute number of DNTs was significantly related with a low number of metastatic sites, better ECOG performance status, BRAF V600 status and normal values of LDH. In addition, the percentage of DNTs was significantly related with normal values of LDH (Table 5).

For the entire population we found that the ORR was significantly related with a lower baseline white blood cell (WBC) count, a higher percentage of lymphocytes, and a lower absolute number of CD3+CD56+ natural killer (NKT) T cells (Table 6). Similarly, OS was negatively related to the WBC count and positively correlated with the percentage of lymphocytes, with correlation coefficients of −0.35 and 0.42, respectively (*p* < 0.05). No correlation was found between PFS and OS, except in patients treated with ipilimumab, for whom OS was correlated with a higher basal value of DNT cells (r = 0.32; *p* < 0.05).

### 3.3. Correlation Between the Change from Baseline T Cells and Therapy Outcomes

We collected a second sample from 47 out of the 68 patients (36 treated with ipilimumab and 11 with PD1 blockade) at the time of their first radiological assessment. Those patients who responded to treatment exhibited a significantly higher percentage of lymphocytes and decreased number of DNTs (Table 7). OS was significantly related to the increase in percentage of CD8+ T cells (r = 0.25, *p* < 0.10).

The trends of DNT cells and of some subpopulations of T cells differed between the group of patients who experienced a response to checkpoint inhibitors and those with progressive disease. In particular, in the group of responsive patients, the median absolute and relative value of DNTs decreased, while the CD4+ and natural killer-like T cells increased (Figure 2a). In patients with unfavorable predictive/prognostic parameters, such as patients with more than three metastatic sites and an LDH over the ULN, we found a statistically significant difference in the change of αβ DNT cells between responsive and non-responsive patients, the latter presenting a large increase in DNT cells (Figure 2b).

In a single case, we observed a particular trend of circulating cells, which enabled us to anticipate a therapeutic outcome. In an ipilimumab-treated patient who developed brain metastases at the first radiological assessment and then underwent brain stereotaxic radiotherapy, we observed an increase of circulating αβ DNT before radiotherapy. After radiotherapy, when a shrinkage of brain lesions and neck lymphadenopathy occurred (Figure 3b) and there was an appearance of vitiligo, we noted a rapid fall in the number of these cells. Moreover, the decrease of αβ DNT cells was paralleled by an increase of CD4+ and NKL, but preceded the reduction in LDH (Figure 3a).

## 4. Discussion

The purpose of this study was to verify whether the blood assessment of T cell subpopulations could help discover novel candidate biomarkers with which to predict the response or resistance of patients to checkpoint inhibitors and to provide a framework for improving melanoma treatment. We focused on a small T subpopulation, the DNT cells, as a potential surrogate for the activation status of the immune system against tumors due to their ability to mirror the degree of both tolerogenic and a cytotoxic activity. Indeed, these cells are depicted as the hinge of a fine balance in the homeostasis of the immune system by clearing autoreactive immune cells as well as regulating allogenic responses [23,24,25,26]. First, we assessed the pre-treatment frequency of T cells and their correlation with clinical features and therapy outcomes. Pre-treatment DNTs were correlated with well-known good prognostic factors such as performance status, LDH, number of metastatic sites, and BRAF V600 status. Moreover, the baseline level of DNTs was correlated with OS of patients treated with ipilimumab, which represented the most common therapy in our population. Previously, it was found that the frequency of these cells was significantly reduced in melanoma as well as haematologic cancers compared to healthy subjects [18,27]. Accordingly, the highest levels of these cells were observed in patients with less severe disease, representing the closest condition to a healthy state. Moreover, the higher value of these cells in our BRAF V600 pretreated patients was in accordance with previous findings that showed an enrichment in DNTs in metastatic biopsies during target therapy [20]. With regard to the baseline predictive features, we found that the ORR, PFS and OS was significantly associated with checkpoint inhibitors and was significantly higher in patients with better ECOG status, an LDH under ULN, and fewer than three metastatic sites, in accordance with the findings of previous studies [28,29,30]. Melanoma at M1a stage was also a factor associated with increased PFS and OS, while the correlation between NRAS mutation and OS was consistent with evidence reporting that this genetic status is linked to better immunotherapy outcomes [31]. Likewise, we found blood cell features that may be predictive biomarkers of a response in patients to checkpoint blockade therapy, for example, a reduced leukocyte count and a higher relative lymphocyte count [2,3,4,5,6,7] were positively correlated with ORR and OS. In our study, the frequency trends of DNTs in the longitudinal analysis revealed that, beyond the expected correlations with increased CD8+ and OS, as well as in whole lymphocytes count and ORR [32], we found that for CD3+ cells, the DNT value was only significantly correlated with values that indicated a patient response. We found a negative relationship between ORR and the number of DNT cells. This evidence may help reveal the exact role of DNTs in the immune system when faced with melanoma. Although studies on the in vivo role of circulating human DNTs in cancer patients are limited, DNTs were found in cytofluorimetric analyses of tissues from many tumor types [19,20,33]. In diverse tumor infiltrates, these cells displayed overlapping features with tolerogenic T cells [20]. An enrichment of DNTs was documented in melanoma-invaded lymph nodes compared to negative lymph nodes. Moreover, a significant increase of DNTs in suppressive/regulatory cells was found in infiltrated lymph nodes of patients whose melanoma had progressed compared to patients who had not progressed, suggesting that these T cells are involved in immune mechanisms triggering metastatic progression [19]. Anticancer therapy could be modified based on the tumor frequencies of these cells [20]. Targeted therapy with BRAF and MEK inhibitors led to an increase of DNTs along with the loss of markers of T cell activation such as CD69 [20]. Such evidence supports the idea of an immunoregulatory role of DNTs, with the ability to kill activated Melan A-HLA2-restricted CD8+ T cells with the same TCR specificity through cell-to-cell contact mediated by Fas/FasL ligand [24] in a feedback loop that limits clonal expansion of alloantigenic-specific T cells and preserves the homeostatic balance of the immune system. Our finding leads to the hypothesis that this cellular mechanism could be exploited by melanoma as a tool to resist checkpoint inhibitors. An in vitro co-culture with melanoma cells caused DNTs to upregulate the expression of immunophenotypic markers of tolerogenic behavior such as CD30 [19]. However, other reports indicated that DNTs are more likely to be ontogenetically related to CD8+ and exert a cytotoxicity towards haematological, lung, melanoma and pancreatic cancer cells [11,12,13,14]. In those reports, autologous and allogenic DNTs were co-cultured in vitro with cancer cell lines and showed cytotoxicity that was mediated by perforine and granzyme B after a cell interaction induced by a high affinity TCR or by the activating immune receptor NKG2D, whose expression may have been modulated by PD1 inhibitors [11,12,13,14]. An anti-cancer, rather than immunosuppressive, role of DNTs could appear counterintuitive but it should be underlined that the in vitro scenario is unlike the complex interactions between the different cell actors involved in the immune response to tumors that are observed in vivo. Moreover, DNTs can exert cytotoxicity on cells other than T cells and can kill B cells [34,35], antigen-presenting cells [36], and NK cells [37] in perforine- and granzyme B-mediated mechanisms. Evidence of the in vivo dynamic interplay between DNTs and other T lymphocyte subpopulations was provided by our longitudinal analysis, during which we observed an opposing trend between DNTs and CD4+, CD3/56+ and NK. Moreover, these trends contrasted in the two different groups of responsive and non-responsive patients. Of interest, the in-treatment decrease of circulating DNTs was correlated with the response and retained its strength in subgroups of patients with poor prognostic factors such as LDH over the ULN and high disease burden. This reduction in circulating DNTs showed potential for the detection of a disease response earlier than the decrease of LDH levels and the appearance of radiological response, as was observed in a single case for a patient who experienced a response after a radiotherapy course (Figure 3).

The main limitation of our study was that we did not investigate the clonality of circulating DNTs in our melanoma population. Thus, we were unable to replicate the in vitro experiments that showed the cytotoxicity of DNTs towards CD8+ or other immune cells. Furthermore, we did not explore the cytokine levels and their influence on the frequency or behavior of DNTs during the course of treatment with checkpoint inhibitors. Finally, the sample size of our population was not sufficient to provide definitive conclusions and patients were treated heterogeneously according to the previous approved guidelines.

## 5. Conclusion

In conclusion, we have shown that the number of DNT cells changes in response to treatment with checkpoint inhibitors and could be adept at sensing a patient’s immune response to melanoma. The similar variation of DNT cells and CD4, as well as other immune actors, may improve the reliability of lymphocyte assessment and warrants further investigation of DNTs as a potential target in checkpoint inhibitor resistant melanoma.

## Figures and Tables

**Figure 1 cells-10-00406-f001:**
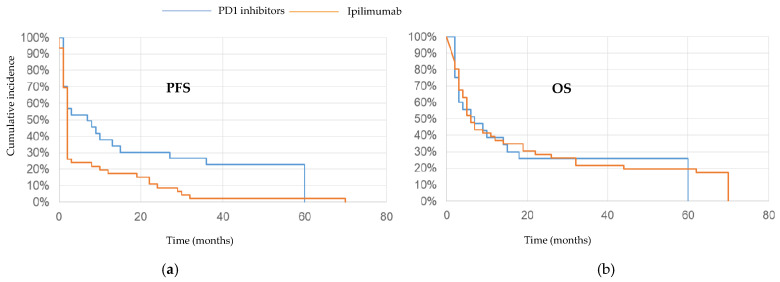
(**a**) Progression-free survival (PFS) of metastatic melanoma patients treated with ipilimumab (orange line) and PD1 inhibitors (blue line); (**b**) overall survival (OS) of the same population treated with the two class of check-point inhibitors.

**Figure 2 cells-10-00406-f002:**
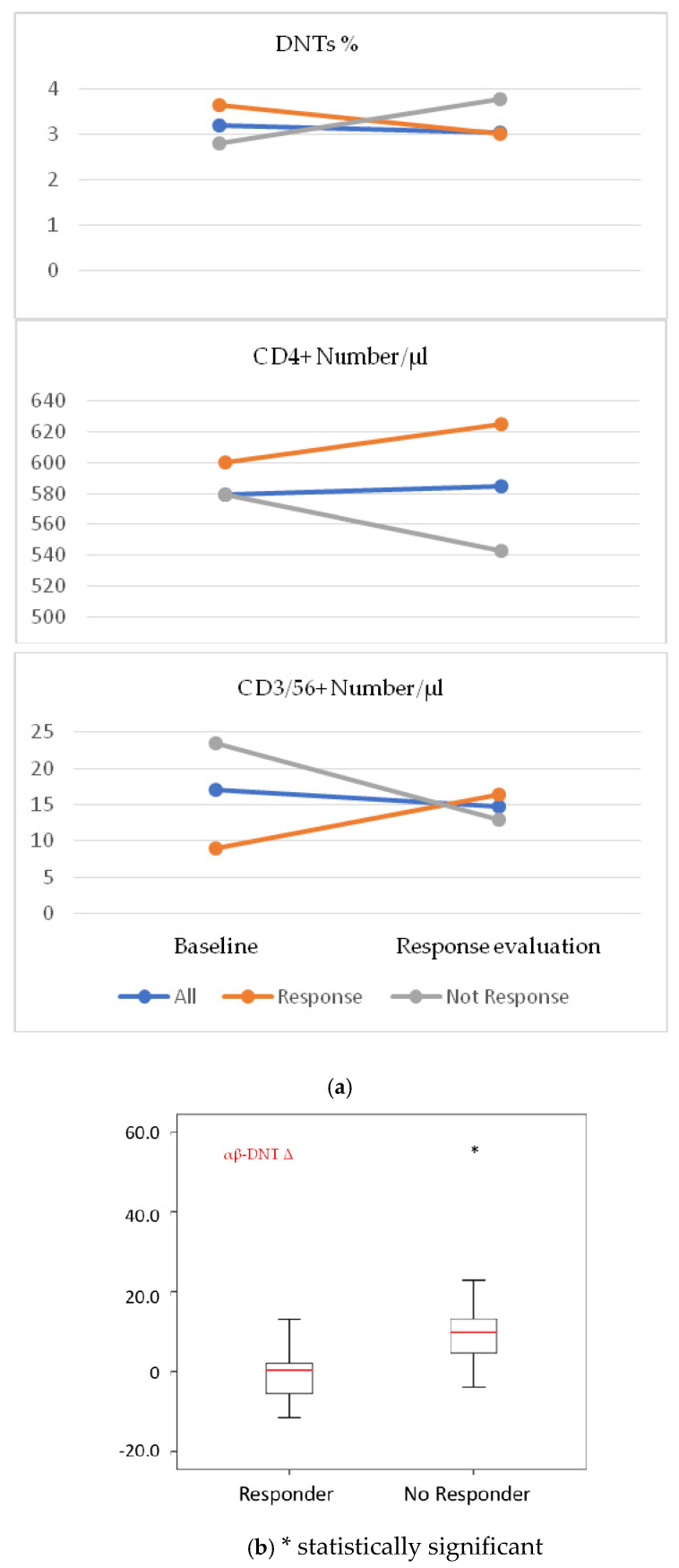
(**a**) Contrasting trends of DNT, CD4+, CD56+ T cells in patients who responded and who did not respond to checkpoint inhibitor therapy. (**b**) Evidence of statistically significant difference between the change in the number of αβ-DNTs between responders and non-responders among patients with baseline LDH > ULN.

**Figure 3 cells-10-00406-f003:**
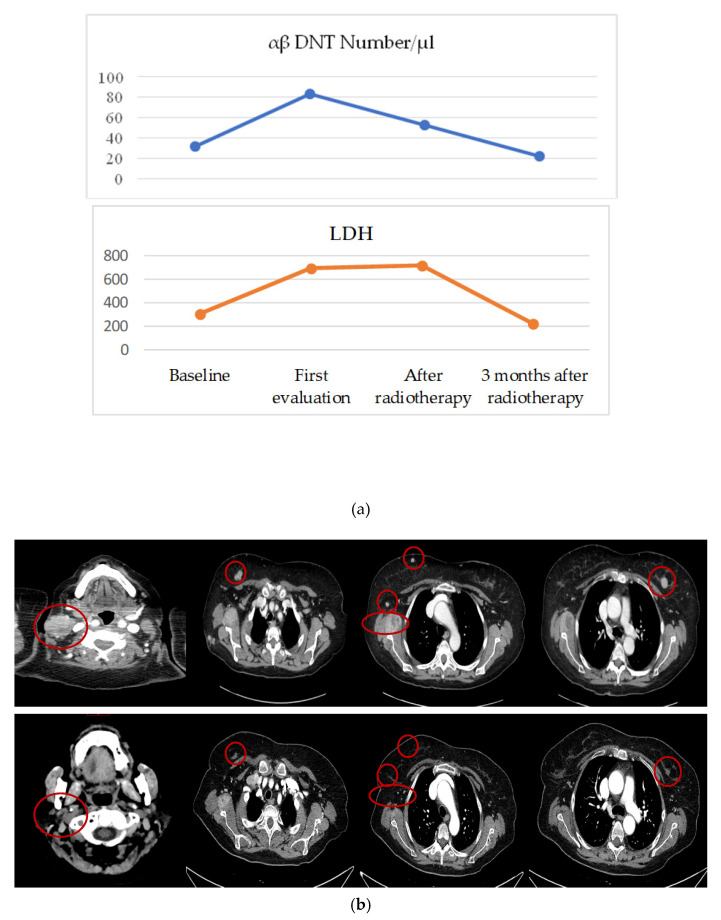
(**a**) Variation of circulating αβ DNTs and LDH in a single patient receiving ipilimumab at baseline and at first radiological assessment, showing the progression of disease, then 1 and 3 months after a course of brain radiotherapy. (**b**) Computer tomography scan of lymph node and soft tissue metastases after the completion of four cycles of ipilimumab (upper part of the panel) and three months after a course of brain radiotherapy (lower part of panel) when a remarkable radiological shrinkage was documented as a result of a conceivable abscopal effect.

**Table 1 cells-10-00406-t001:** Baseline features of the metastatic melanoma population treated with checkpoint inhibitors.

Feature	% (n = 68)
Age	
Median (interquartile range) years	65 (34–85)
Sex	
Female	43 (29)
Male	57 (39)
Type of melanoma	
Cutaneous	79 (54)
Mucosal	3 (2)
Uveal	4 (3)
Unknown origin	14 (9)
BRAF V600 mutation	53 (36)
NRAS Q61 mutation	10 (7)
TNM stage at diagnosis	
II	44 (30)
III	38 (26)
IV	18 (12)
Median Disease Free Survival (months)	19 (0–144)
M stage	
M1a	21 (14)
M1b	14 (10)
M1c	47 (32)
M1d	18 (12)
Number of metastatic sites	
< 3	41 (28)
≥3	59 (40)
ECOG	
0	31 (21)
1	48 (33)
2	21 (14)
LDH	
<ULN *	47 (32)
≥ULN	53 (36)
Line of systemic treatment	
Ipilimumab cohort	
As first line	40 (19)
As second line	60 (29)
PD1 inhibitors cohort	
As second line	50 (10)
As third line	50 (10)

* ULN: Upper Limit of Normal.

**Table 2 cells-10-00406-t002:** Clinical features with respect to checkpoint inhibitor therapy for Overall Response Rate.

ECOG ^(^**^)^					
	0	1		2	Total
CR/PR	10 (58.8%)	7 (41.2%)		0 (0%)	17 (100%)
PD	8 (18.6%)	21 (48.8%)		14 (32.6%)	43 (100%)
Total	18 (30.0%)	28 (46.7%)		14 (23.3%)	60 (100%)
**Number of Metastatic Sites ^(^*^)^**					
CR/PR	< 3		≥3		Total
	9 (52.9%)		8 (47.1%)		17 (100%)
PD	13 (30.2%)		30 (69.8%)		43 (100%)
Total	22 (36.7%)		38 (63.3%)		60 (100%)
**Age ^(^*^)^**					
CR/PR	72 (64–76)				
PD	64 (54–72)				

* *p*-value < 0.10; ** *p*-value < 0.05.

**Table 3 cells-10-00406-t003:** Clinical features with respect to checkpoint inhibitor therapy for PFS. PFS values were summarized in terms of median and interquartile range (1st and 3rd quartiles).

Features	PFS (Months)
**M** **Stage**	
M1a	8 (3–30)
M1b	3 (3–36.5)
M1c	3 (2–3)
M1d	2 (2–3)
**ECOG**	
0	9 (3–32)
1	3 (2–5)
2	2 (2–3)
**LDH**	
<ULN	3 (3–27)
≥ULN	3 (2–3)
**Number** **of Metastatic Sites**	
< 3	3 (3–24)
≥3	3 (2–3)

**Table 4 cells-10-00406-t004:** Clinical features with respect to checkpoint inhibitor therapy for OS. OS values were summarized in terms of median and interquartile range (1st and 3rd quartiles).

Features	OS (Months)
**M Stage**	
M1a	48 (4–63)
M1b	16.6 (6.5–62.5)
M1c	6 (3–32)
M1d	4 (2–6)
**ECOG**	
0	41 (7–70)
1	7 (4–29)
2	3 (2–4)
**LDH**	
<ULN	10 (6–62)
≥ULN	4 (3–18)
**Number of Metastatic Sites**	
< 3	12.5 (4–62)
≥3	6 (3–22)
**Mutation Status**	
BRAF/NRAS wt	6 (3.5–20.5)
BRAF V600	6 (3–12)
NRAS Q61	60 (48–62)

**Table 5 cells-10-00406-t005:** DNT baseline value distribution with respect to clinical features in the metastatic melanoma population treated with checkpoint inhibitors. DNT values were summarized in terms of median and interquartile range (1st and 3rd quartiles).

Features	DNT Absolute Number/µL	DNT Percentage among Lymphocytes	DNT Percentage among CD3+ Cells
**Mutational status**	(*)		
BRAF/NRAS wild type	29.27 (19.17–50.82)		
BRAF V600	65.69 (23.32–101.33)		
NRAS Q61	37.64 (26.49–101.19)		
**LDH**	(**)	(**)	(**)
<ULN	48.05 (23.46–106.99)	3.59 (1.94–7.06)	5.20 (3.33–10.05)
≥ULN	28.57 (19.26–62.68)	2.22 (1.44–4.01)	3.13 (2.06–5.65)
**ECOG**	(**)		
0	68.47 (28.57–106.99)		
1	30.58 (19.24–80.74)		
2	26.51 (9.57–30.90)		
**Number of metastatic sites**	(*)		
< 3	35.81 (101.34–169.32)		
≥3	28.65 (71.09–167.90)		

* *p*-value < 0.10; ** *p*-value < 0.05.

**Table 6 cells-10-00406-t006:** White blood cell baseline values in relation to ORR clinical outcomes of checkpoint inhibitors. Values according to clinical outcomes were summarized in terms of median and interquartile range (1st and 3rd quartiles).

Overall Response Rate			
	WBC/µL	%LY	CD3/56%
CR/PR	6390 (5670–6850)	22.68 (18.91–26.16)	0.6 (0.2–2.5)
PD	8158 (5680–11,450)	15.4 (12.43–21.54)	1.5 (0.43–4.1)

**Table 7 cells-10-00406-t007:** Distribution of delta (Δ) of blood cell variation with respect to ORR clinical outcomes of checkpoint inhibitors.

Overall Response Rate		
	%LY Δ (*)	Absolute Number DNT Δ (*)
CR/PR	−0.87 (−5.70–6.74)	2.02 (−25.86–23.24)
PD	3.13 (−0.04–7.80)	−0.43 (−11.91–9.18)

* *p*-value < 0.10

## Data Availability

All data generated or analyzed during this study are included in this published article**.**

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
