# Peer review of "Examining the Relationship between Circulating CD4− CD8− Double-Negative T Cells and Outcomes of Immuno-Checkpoint Inhibitor Therapy—Looking for Biomarkers and Therapeutic Targets in Metastatic Melanoma"

_cells, 2021, doi:10.3390/cells10020406_

Round 1

Reviewer 1 Report

MAJOR POINTS

As stated by Authors, the ambitious aim of the study is to discover novel useful candidate biomarkers to predict the response or resistance to checkpoint inhibitors and to provide a framework for improving melanoma treatments. In the light of this, sample size must be increased, since patients enrolled are 68 only (even divided into two groups based on therapeutic treatment received).

Authors reported that targeted therapies could influence/modify frequences of DNT cells. According to this consideration, how can these cells be also informative/predictive of treated patients response?

Many speculative hypothses, raised within dicussion, would deserve to be confirmed by subsequent experimetal studies. However, Authors state that they are “unable to replicate the in vitro experiments” due to the lack of investigation about “the clonality of circulating DNTs”. This represents a fundamental concern that markedly limit the power/potential of the study.

Extensive revision concerning english grammar and style is absolutely needed.

MINOR POINTS

INTRODUCTION

Text from line 37 to line 50 should be shortened/deleted, since it does not focus on the main scientific issue.

FIGURES AND TABLES

Quality of Figure 1 should be improved. Graphs and images reported in Figure 2 and 3 are too small and obstacle reader’s comprehention of the results. Table 2 should be presented as text file, not as a picture. Table 5 should be scaled down.

Author Response

Point 1.

As stated by Authors, the ambitious aim of the study is to discover novel useful candidate biomarkers to predict the response or resistance to checkpoint inhibitors and to provide a framework for improving melanoma treatments. In the light of this, sample size must be increased, since patients enrolled are 68 only (even divided into two groups based on therapeutic treatment received).

Response: The reviewer comments is proper. We are unable to widen the sample size of population. We added a statement to highlight this weakness as a limitation of our study in the discussion  section (see line 309)

Point 2.

Authors reported that targeted therapies could influence/modify frequences of DNT cells. According to this consideration, how can these cells be also informative/predictive of treated patients response?

Response: We found that a higher absolute number of DNTs was significantly related with BRAF V600 status at baseline, before the beginning of checkpoint inhibitors. We investigated the predictive role of the circulating DNTs in patients treated with checkpoint inhibitors. All our BRAF V600 mutated patients, excepting one, were previously treated with BRAF/MEK inhibitors and all these patients started checkpoint inhibitors after the onset of a resistance to target therapy. Moreover, all these patients experienced a partial response as best response during the previous treatment with target therapy. Thus, our study was unable to assess how DNTs could be associated to response to target therapy. In the discussion section we highlighted that an enrichment in DNTs in metastatic biopsies during target therapy was previous reported by Greenplate, A.R (reference 20) but also these authors did not report if this increase in DNTs was correlated or not to a response.

Point 3.

Many speculative hypothses, raised within dicussion, would deserve to be confirmed by subsequent experimetal studies. However, Authors state that they are “unable to replicate the in vitro experiments” due to the lack of investigation about “the clonality of circulating DNTs”. This represents a fundamental concern that markedly limit the power/potential of the study.

Response: As the reviewer reported, we clearly stated that we decided not to perform in vitro experiments mainly due to the lack of investigation about the clonality of DNTs. In the discussion section, we also specify that DNTs usually work in a complex context within the immune system and their role is coordinated with multiple cellular actors (T cells, B cells, Natural Killer Cells and Antigen Presenting Cells). Thus, this scenario is hardly reproducible in vitro and we stated that  our study represented an exploratory investigation in a patient population. Our data could be validated in a larger prospective study and could open the road to further explore the role of these cells in the mechanisms of resistance to immunotherapy.

Point 4.

Extensive revision concerning english grammar and style is absolutely needed.

Response: We proofread the manuscript by the professional editing service of MDPI.

Point 5.

Text from line 37 to line 50 should be shortened/deleted, since it does not focus on the main scientific issue.

Response: As the reviewer suggested, we shortened this sentences in the introduction section.

Point 6.

Quality of Figure 1 should be improved. Graphs and images reported in Figure 2 and 3 are too small and obstacle reader’s comprehention of the results. Table 2 should be presented as text file, not as a picture. Table 5 should be scaled down.

Response: We tried to improved the quality of figure 1. We magnified and enlarged the figure 2 and 3 and the relative graphics. We presented the table 2 as text file. We adjusted table 5 position.

Reviewer 2 Report

Dear authors,

This manuscript is hard to read and understand and needs an extensive editing of English language and style. Moreover, the figures are not well presented:  

-Improve figures: Kaplan-Meier plots in figure 1 are blurred and hard to read, Table 2 seems to be converted into a picture, I would hardly recommend keeping the formatting of tables comparable. Figure 2 is not well organized and impossible to read, please increase the font size of labeling and numbers of scales and keep the size of panels comparable. In Figure 3 the picture of the patient is useless as you cannot see more as depicted by MRTs. So I would suggest to remove this. 

Author Response

Point 1

This manuscript is hard to read and understand and needs an extensive editing of English language and style.

Response 1: We proofread the manuscript by the professional editing service of MDPI.

Point 2: Moreover, the figures are not well presented:  -Improve figures: Kaplan-Meier plots in figure 1 are blurred and hard to read, Table 2 seems to be converted into a picture, I would hardly recommend keeping the formatting of tables comparable. Figure 2 is not well organized and impossible to read, please increase the font size of labeling and numbers of scales and keep the size of panels comparable. In Figure 3 the picture of the patient is useless as you cannot see more as depicted by MRTs. So I would suggest to remove this. 

Response: As the reviewer suggested we improved the quality of figure 1. We presented the table 2 as text file and made the formatting of tables comparable. We magnified and enlarged the figure 2 and the relative graphics. We removed the picture of patient whose aim was to show the vitiligo-like skin depigmentation as described in the text and in the figure legend.

Round 2

Reviewer 1 Report

Quality and organization of both Figures and Tables must be improved.

Author Response

Response to Reviewer 1

Point 1. Quality and organization of both Figures and Tables must be improved.

Response 1: We redesigned the figures. We also attached the power point version of these figures to provide a better viewable format. Moreover, we splitted table 2 in table 2, 3, and 4 (related to ORR, PFS, OS respectively) and renamed the other tables in the manuscript.

Response to Reviewer 1

Point 1. Quality and organization of both Figures and Tables must be improved.

Response 1: We redesigned the figures. We also attached the power point version of these figures to provide a better viewable format. Moreover, we splitted table 2 in table 2, 3, and 4 (related to ORR, PFS, OS respectively) and renamed the other tables in the manuscript.

Reviewer 2 Report

In the present manuscript, others have investigated the role of CD4/CD8 double negative T cells (DNTs) in response to and in the treatment of melanoma patients with immune checkpoint inihibors.

The manuscript has been improved, however the labeling in figures 2 and 3 are still not readable in the printout/pdf at a resolution of 100%

The general major concerns are indeed how DNTs functionally respond to therapeutics and by their interaction with melanoma cells. It´s hard to understand why DNTs are at a high level in heatlhy individuals, decrease in melanoma patients but are important for a therapeutic responce. It would be worthwhile to investigate sections of melanoma metastases for the presence of the different T cell subsets and levels of PD-L1. To gain insight into mechanisms of how DNTs are indeed regulated and respond to immune checkpoint inhibiors a transcriptome profiling of DNTs would be interesting as well.

Author Response

Response to Reviewer 2

Point 1: The manuscript has been improved, however the labeling in figures 2 and 3 are still not readable in the printout/pdf at a resolution of 100%

Response 1: We redesigned the figures. We also attached the power point version of these figures to provide a better viewable format.

Point 2: The general major concerns are indeed how DNTs functionally respond to therapeutics and by their interaction with melanoma cells. It´s hard to understand why DNTs are at a high level in heatlhy individuals, decrease in melanoma patients but are important for a therapeutic responce. It would be worthwhile to investigate sections of melanoma metastases for the presence of the different T cell subsets and levels of PD-L1. To gain insight into mechanisms of how DNTs are indeed regulated and respond to immune checkpoint inhibiors a transcriptome profiling of DNTs would be interesting as well.

Response 2: The aim of this study was to examine whether the blood evaluation of T cell subpopulations could help predict or assess melanoma response to checkpoint inhibitors. We focused on DNTs because these cells were previously reported to show in vitro cancer cytotoxicity (see references 11 to 14), while other reports documented the presence of these cells in soft tissue and lymph nodes metastases of melanoma patients in shape of immunosuppressive cells (see references 19 and 20). The higher blood value of DNTs in healthy subjects compared to cancer patients, and the presence among patients of the positive correlation between DNTs and normal LDH, better ECOG and lower number of metastases suggested us the hypothesis that these cells are involved in the homeostasis of the immune system. Therefore, the alterations induced by melanoma itself and the extent of these alterations (LDH, ECOG and tumor load are surrogates of this extent) may support our hypothesis. Moreover, owing to the ability of DNTs to limit clonal expansion of alloantigenic-specific T cells, we hypotized that this cellular mechanism could be exploited by melanoma as a resistance mechanism to checkpoint inhibitors because in responder patients we found a decrease of the blood value of DNTs, as opposed to the increase of DNTs that was observed in non-responders. All these considerations were reported in the discussion section (see lines 279 to 284).

We decided to use blood samples instead of FFPE tissue because the assessment of DNTs is done by flow cytometer evaluation in fresh tissues (see reference 19, 20 and also Stankovic B, Bjørhovde HAK, Skarshaug R, Aamodt H, Frafjord A, Müller E, Hammarström C, Beraki K, Bækkevold ES, Woldbæk PR, et al. Immune Cell Composition in Human Non-small Cell Lung Cancer. Front Immunol. 2019 Feb 1;9:3101. doi: 10.3389/fimmu.2018.03101). Thus, blood samples allowed us to expand our exploratory cohort to an adequate number of patients without performing invasive procedures as biopsies in less accessible metastatic sites.

As suggested by reviewers, it would be wortwhile to perform a trascriptome profiling of DNTs, to provide valuable informations on the immunological mechanism underlying the observed blood trends. We were unable to perform these analyses immediately, hence we already planned to prospectively collect blood DNTs from melanoma patients and to assess their TCR clonality and their gene profile to confirm the evidences of the present exploratory study.